# Concurrent Gene Insertion, Deletion, and Inversion during the Construction of a Novel Attenuated BoHV-1 Using CRISPR/Cas9 Genome Editing

**DOI:** 10.3390/vetsci9040166

**Published:** 2022-03-30

**Authors:** Chun-Yu Liu, Ming Jin, Hao Guo, Hong-Zhe Zhao, Li-Na Hou, Yang Yang, Yong-Jun Wen, Feng-Xue Wang

**Affiliations:** 1College of Veterinary Medicine, Inner Mongolia Agricultural University, Hohhot 010018, China; chunyuliu_vet@163.com (C.-Y.L.); jming1993@126.com (M.J.); gh19960627@126.com (H.G.); hongzhezhao@163.com (H.-Z.Z.); hou163@126.com (L.-N.H.); 2School of Life Sciences, Inner Mongolia University, Hohhot 010018, China; yang55797961@163.com

**Keywords:** BoHV-1, CRISPR/Cas9, homologous recombination, insertion, inversion

## Abstract

Bovine herpesvirus type I (BoHV-1) is an important pathogen that causes respiratory disease in bovines. The disease is prevalent worldwide, causing huge economic losses to the cattle industry. Gene-deficient vaccines with immunological markers to distinguish them from wild-type infections have become a mainstream in vaccine research and development. In order to knock out the gE gene BoHV-1, we employed the CRISPR/Cas9 system. Interesting phenomena were observed at the single guide RNA (sgRNA) splicing site, including gene insertion, gene deletion, and the inversion of 5′ and 3′ ends of the sgRNA splicing site. In addition to the deletion of the gE gene, the US9 gene, and the non-coding regions of gE and US9, it was found that the US4 sequence, US6 sequence, and part of the US7 sequence were inserted into the EGFP sgRNA splicing site and the 3′ end of the EGFP sequence was deleted. Similar to the BoHV-1 parent, the BoHV-1 mutants induced high neutralizing antibodies titer levels in mice. In summary, we developed a series of recombinant gE-deletion BoHV-1 samples using the CRISPR/Cas9 gene editing system. The mutant viruses with EGFP^+^ or EGFP^−^ will lay the foundation for research on BoHV-1 and vaccine development in the future.

## 1. Introduction

Bovine herpesvirus type I (BoHV-1) can cause infectious bovine rhinotracheitis (IBR) and infectious pustular vulvovaginitis (IPV), which are caused by BoHV-1.1 and BoHV-1.2 [1,2,3], respectively. The disease is prevalent worldwide, causing huge economic losses in the cattle industry. High fever, decreased milk production, purulent nasal fluid, nasal redness (red nose disease), and conjunctivitis are the most common clinical symptoms of BoHV-infected cattle. Furthermore, BoHV-1 belongs to the family *Herpesvirus*, the subfamily *Herpesvirus*, and the genus *Varicellovirus* [2,4]. Notably, BoHV-1 only infects cattle. It can establish a latent infection in the sensory neurons of the trigeminal ganglion (TG) under a strong immune response. The latent infection of BoHV-1 makes disease control more difficult. Therefore, it is necessary to develop new vaccines that can distinguish between wild virus infection and vaccine immunity. The gene-deficient vaccines have immunological markers and the deleted genes can be used for the differential diagnosis of wild-type infection and vaccine immunization. Related studies have reported that the BoHV-1 Ge gene will not be detoxified through the nasal cavity even if it is reactivated after deletion [5,6]. Deletion of the Ge gene reduces viral neurotropism, thereby reducing latency and risk of reactivation. Additionally, BoHV-1-ΔgE marker vaccine is safer than the MLV because it is not transmitted from vaccinated to non-vaccinated animals, rarely shed following latency reactivation, and vaccinated animals are distinguishable from infected animals [7]. At present, research on the prevention and treatment of latent human herpesvirus infection mainly focuses on the following two points: the first is vaccination before primary infection, and the other is the early application of antiviral drugs. Relevant studies have shown that 90% of people have latent infection with herpesvirus, and the peak age of forming this infection is in early childhood [8,9,10]. Therefore, early immunization to prevent latent infection of herpesvirus is considered to be the most likely effective method to prevent the latent infection of trigeminal ganglion. Live attenuated vaccines allow the virus to establish a life-long incubation period. Furthermore, most live attenuated vaccines cause immunosuppression in calves whose immune systems are not fully developed, resulting in disease. Additionally, when the vaccine strain is replicated in the same bovine, there is a risk of strong virulence. Inactivated vaccine or subunit vaccine is one of the ways by which to avoid latent infection.

The BoHV-1 viral genome is double-stranded DNA, encoding about 70 proteins, of which 33 structural proteins and more than 15 nonstructural proteins have been identified [11]. The genome is 135–140 kb long. It consists of a unique long sequence (UL), a unique short sequence (US), and repetitive intermediate repeat (IRS) and terminal repeat (TRS) sequences flanking the US region [12]. During DNA replication, the UL and US regions are relatively flipped (i.e., UL–US to US–UL), which can generate two isomeric genomes [13].

The incubation period and periodic reactivation of BoHV-1 can cause the virus or virus particles to travel anteriorly to the primary infection site at the end of the axon and infect the epithelial cells of the nasopharynx and eye, where the virus replicates and falls off. The glycoprotein E (gE) and Us9 homologues of BoHV-1 are essential for viral anterograde neuron transport in primary neurons in vitro [14]. The anterograde neuron transport of BoHV-1 from the trigeminal ganglion (TG) to the nose and eyes requires the gE gene. The BoHV-1 gE gene deletion virus can be reactivated from the incubation period after infecting calves, but it cannot be transported anteriorly from the TG to the nerve endings of the nose or cornea. This shows that the gE gene is one of the determinants of BoHV-1 virulence and anterograde neuron transport function [5]. Furthermore, BoHV-1 expresses two membrane proteins, gE/gI, which play a key role in axon anterograde transport and cell-to-cell transmission [15]. The gE gene is a recognized virulence factor for all known members of the *Alphaherpesvirinae* subfamily [6,16]. The vast majority of gE in infected cells combine with gI to form heterodimers and play an important role in cell-to-cell transmission [17,18]. In fact, the deletion of viral gE will reduce the expression of gE/gI. Deleting gE/gI-related proteins may affect viral anterograde axon transport and may also increase protein expression and phosphorylation. A highly attenuated BoHV-1 strain, which is marked as gE gene deletion [19], is used for vaccine preparation in the BoHV-1 prevention and control plan of some EU countries.

Mammalian cells have evolved complex repair mechanisms to prevent genomic instability. Improperly repaired double-strand breaks (DSBs) can cause chromosome loss or potentially carcinogenic chromosomal rearrangements [20]. Cells have powerful repair pathways, including non-homologous end joining (NHEJ) and homologous recombination (HR) [21]. In the presence of homologous donor DNA, HR is used to repair damaged DNA. NHEJ is an error-prone mechanism that causes sequence insertions and deletions [22]. Among mammalian cells, either single-strand breaks or DSBs induce HR repair mechanisms [23]. Moreover, HR is the main DSB repair mechanism in mammalian cells [24]. The discovery of a genome DNA repair mechanism—clustered regularly interspaced short palindromic repeats (CRISPR) and CRISPR-associated nucleases (CRISPR/Cas)—by Mojica and Jansen in 2002 led to the CRISPR/Cas9 system becoming a powerful tool for DNA editing [25]. This method can successfully edit the genome DNA of cells and animals [26]. It also can be used to manipulate large viral genomes, such as herpes simplex virus type I (HSV-1) [27], adenovirus (ADV) [28], pseudorabies virus (PRV) [29], Cowpox virus (CPV) [30], Epstein–Barr virus (EBV) [31], cytomegalovirus (CMV) [32], hepatitis B virus (HBV) [33], duck plague virus (DPV) [34], and Marek’s disease virus (MDV) [35].

In the present study, we constructed a gene deletion recombinant virus of BoHV-1 with the CRISPR/Cas9 system. The reporter gene EGFP was used to recombine and replace the gE gene of BoHV-1. Finally, we obtained a series of recombinant viruses via CRISPR/Cas9 editing. They all have similar growth characteristics and a similar plaque morphology to the parental BoHV-1. Some phenomena appeared in the EGFP reporter gene sequence recombined at the position of the gE gene, including gene insertion, deletion, and an inversion of the 5′ and 3′ ends of the sgRNA cleavage site. However, these changes did not affect virus survival.

This phenomenon occurs when the non-coding region between the BoHV-1 gE gene and the US9 gene, the gE gene, and part of the US9 sequence are deleted by the designed donor plasmid. The US4, US6, and part of the US7 gene are inserted into the sgRNA splicing sites of the EGFP recombined in the gE position of the viral genome. This results in the loss of the EGFP 3′ end sequence. We found that the six recombined BoHV-1 samples have similar growth characteristics and immunogenicity to the parental BoHV-1. This indicates that the BoHV-1 gE/EGFP^−^ obtained via CRISPR/Cas9 editing provides a basis for the development of BoHV-1 gene deletion vaccines and BoHV-1 as a delivery vector.

## 2. Materials and Methods

### 2.1. Viruses and Cell Lines

Parental BoHV-1 was isolated, identified, and stored in our Lab. Madin-Darby bovine kidney cells (MDBK), baby hamster kidney cells (BHK-21), and human embryonic kidney cells (HEK293T) were purchased from Kunming Cell Bank. Primary bovine testis cells (BT) and African green monkey kidney cells (VERO E6) were donated by Harbin Veterinary Research Institute. MDBK, BHK-21, HEK293T, Vero E6, and BT cells were cultured in Dulbecco’s modified Eagle’s medium (Gibco, Grand Island, NY, USA). All media were supplemented with 10% heat-inactivated fetal bovine serum.

### 2.2. Construction of sgRNA and Donor Plasmids

All the primers and sgRNAs were designed based on the gE gene of the BoHV-1 genome (GenBank: AJ004801.1) and the EGFP gene (GenBank: MN153298.1). The PCR primers targeting the gE gene or EGFP were designed with Primer 5.0 software. Four pairs of sgRNAs of BoHV-1 gE and three pairs of sgRNA of EGFP were developed using E-CRISP-Version 5.4 online software (http://www.e-crisp.org/E-CRISP/) (accessed on 27 January 2022). The primers and sgRNAs were synthesized in BGI. The sgRNAs were denatured, annealed, extended, and cloned into the px459 vector (Addgene) to obtain px459-gE-sgRNA1, px459-gE-sgRNA2, px459-gE-sgRNA3, and px459-gE-sgRNA4. In the same way, we obtained the following three plasmids targeting the EGFP gene: px459-EGFP-sgRNA1, px459-EGFP-sgRNA2, and px459-EGFP-sgRNA3. The EGFP gene was also amplified by PCR and cloned into the PCDNA3.1 vector using *KpnI* and *BamHI* to obtain PCDNA3.1-EGFP. The left homology arms (LgE) were amplified with the primer pair LgE-F/R. In order to ensure that the gE gene, the non-coding region between the gE gene and the US9 gene, and the partial US9 gene sequence were deleted simultaneously, we designed three primer pairs, R1gE-F/R, R2gE-F/R, and R3gE-F/R, to amplify the three right homology arms. Then, LgE and RgE were simultaneously cloned into the PCDNA3.1-EGFP and PCDNA3.1 vectors, respectively. The donor vectors were PCDNA3.1-LgE-EGFP-R1gE, PCDNA3.1-LgE-R2gE, and PCDNA3.1-LgE-R3gE. All the plasmids were identified by restriction endonuclease analysis and sequencing. The primers used in this study are listed in Table 1.

### 2.3. Screening of Cell Lines for Gene Editing

MDBK, BT, BHK-21, HEK293T, and VERO E6 cells were cultured in 12-well plates until 80% confluence. A total of 2 μg pf plasmid px458 was transfected into each cell line using Lipofectamine LTX and Plus Reagent (Invitrogen Thermo Fisher Scientific, Waltham, MA, USA). The fluorescence was observed with a fluorescence microscope (ZEISS, HAL100, Jena, Germany) after 48 h. The cells were collected and analyzed with a BD Accuri C6 Plus flow cytometer (BD AccuriTM C6). MDBK, BT, BHK-21, HEK293T, and VERO E6 cells were infected with 10-fold serial dilutions of the virus. Plaques were counted and virus titers were calculated as plaque-forming units (PFU/mL). The experiments were performed three time independently. Based on the above results, a cell line that efficiently edits BoHV-1 was screened.

### 2.4. The Verification of Sgrna Editing Efficiency on Bohv-1

The VERO E6 cells were cultured in 12-well plates until they reached 80% confluence. Two microgram plasmid px459-gE-sgRNA1, px459-gE-sgRNA2, px459-gE-sgRNA3, px459-gE-sgRNA4, and those of the negative control were transfected using Lipofectamine LTX and Plus Reagent, respectively. The cell culture medium was replaced at 6 h after transfection. BoHV-1 (MOI = 1) was inoculated at 12 h after transfection. The viruses were collected 72 h after inoculation. Then, the 10-fold serial dilutions of virus were incubated on 80% of confluent MDBK cells in 12-well plates for plaque testing. The cells were fixed with 10% formaldehyde at room temperature for 4 h 5 days post-infection (pi). The cells were washed three times with phosphate-buffered saline (PBS) and stained with 8% crystal violet for 30 min. Then, the cells were washed and dried. Plaque numbers were recorded and virus titers were calculated (PFU/mL). The experiments were performed three times independently.

### 2.5. Knock Out BoHV-1 gE Gene by CRISPR/Cas9

The CRISPR/Cas9 vectors px459-gE-sgRNA1, px459-gE-sgRNA2, and the linearized donor plasmid (PCDNA3.1-LgE-EGFP-R1gE) were co-transfected into VERO E6 cells at a ratio of 0.5:0.5:1, respectively. The 2 μg px459 plasmid was transfected as the control. The cell culture medium was replaced after 6 h. After 12 h, BoHV-1 (MOI = 1) was inoculated, incubated for 2 h, and then the cells were placed in Dulbecco’s modified Eagle’s medium containing 1% low melting-point agarose, 1% fetal equine serum, and 1% penicillin-streptomycin. After 72 h, the recombinant virus was selected using an inverted fluorescence microscope and further cloned and reproduced in MDBK cells. In the same way, the CRISPR/Cas9 vector px459-EGFP-sgRNA1 and px459-EGFP-sgRNA2 was used to reverse screening for BoHV-1 gE/EGFP^−^. For the single sgRNA group, the linearized donor plasmid (PCDNA3.1-LgE-EGFP-R3gE) was co-transfected with px459-EGFP-sgRNA1/px459 and px459-EGFP-sgRNA2/px459. For the double sgRNA group, the linearized donor plasmid (PCDNA3.1-LgE-EGFP-R3gE) was co-transfected with px459-EGFP-sgRNA1/px459-EGFP-sgRNA2.

### 2.6. Identification of Recombinant Viruses

To confirm the successful recombination of BoHV-1 gE/EGFP^+^, BoHV-1 gE/EGFP^−^, and BoHV-1 gE/US9/EGFP^−^, the viruses were inoculated into MDBK cells for proliferation after plaque clone purification for six generations. The viral genome was extracted using TIANamp Virus DNA/RNA Kit (Tiangen Biotech Co., Ltd., Beijing, China). Using BoHV-gE F/R primers for PCR. The PCR products were sequenced in BGI (Beijing, China) and analyzed using SnapGene and MEGA6 software. The correct recombinant virus was purified by plaque cloning and propagated 10 times to obtain gE-deletion BoHV-1 strains.

### 2.7. Determination of Recombinant Virus Replication Kinetics

The MDBK cells were inoculated with parental BoHV-1 and recombinant viruses (MOI = 0.01). Then, viruses were collected at 12, 24, 36, 48, 60, and 72 hpi. After freezing and thawing the infected cells three times, the cell supernatant was collected and incubated with MDBK cells in 96-well plates with 10-fold serial dilutions. The viruses were infected with 0.1 mL/well. The cytopathic effect was observed daily. The tissue culture infective dose (TCID_50_) of each recombinant virus was determined using the Reed–Muench method at 4 dpi. All data are shown as the average of three independent experiments.

### 2.8. Off-Target Detection and Morphological Verification of Virus Particles

Parts of the BoHV-1 gene (UL22, UL23, UL27, UL44, UL47, UL53, US4, US6, US7, US8, and US9) were selected to design primers (Table 1). PCR and sequencing were used to verify whether the mutant BoHV-1 generated by the CRISPR/Cas9 tool was off-target shearing in the rest of the gene. At the same time, in order to determine whether the morphology of the mutant BoHV-1 had changed, the BoHV-1, BoHV-1 gE/EGFP^−^, BoHV-1 one, BoHV-1 two, BoHV-1 three, BoHV-1 four, and BoHV-1 five were purified by a sucrose density gradient ultracentrifugation or filtered and concentrated using a 3000 MWCO Amicon Ultra-15 centrifugal filter (EMD Millipore, Tullagreen, Ireland). The concentrated virus was adsorbed onto a copper net for 5 min, and the excess virus was absorbed with filter paper. The concentrated virus was then dyed with 2% phosphotungstic acid (pH 7.0) for 30 s and the excess dye solution was absorbed with filter paper. The morphology of the virus was observed using a JEOL 1230 transmission electron microscope (JEOL, Tokyo, Japan).

### 2.9. Mouse Immunization and Determination of Virus Neutralizing Antibodies

Forty 6–8-week-old female BALB/c mice (Charles River Laboratories, Beijing, China) were randomly divided into 8 groups (5 mice in each group). On 0 d and 21 d, 100 μL of BoHV-1, BoHV-1 gE/EGFP^−^, BoHV-1 one, BoHV-1 two, BoHV-1 three, BoHV-1 four, and BoHV-1 five with a titer of 10^8^ TCID50/mL were injected intramuscularly. Every 7 d after the initial immunization, blood was collected from the heart to separate serum for the virus neutralization test. Serum samples were inactivated at 56 °C for 30 min, serial two-fold dilution was conducted in DMEM, and an equal volume of 200 TCID_50_ diluted BoHV-1 was added to a final volume of 200 μL and kept at 37 °C in 5% CO_2_ for 1 h. MDBK cells were inoculated with 100 μL of neutralization mixture at 37 °C in 5% CO_2_ for 2 h. Each sample was replicated four times. The mixture was removed and DMEM containing 2% fetal equine serum was added. The mixture was cultured at 37 °C in 5% CO_2_ for 5 days. The cytopathic effect (CPE) was observed over time, and the 50% neutralization titer of the serum was calculated using the Reed–Muench method.

### 2.10. Statistical Analysis

The experimental data are expressed as mean ± standard deviation. The differences were analyzed using the ANOVA test. *p* < 0.05 was considered significant. For samples that lacked normality, we used the Wilcoxon Rank Sum Test. For samples that conform to normality, we employed an ANOVA 1-way analysis with Tukey’s Post hoc test for individual comparisons when equal variance was satisfied. In cases where equal variance was not satisfied, we used the Kruskal–Wallis test.

## 3. Results

### 3.1. The Construction of Donor Plasmids

Three BoHV-1 gE-specific CRISPR/Cas9 samples were designed and constructed successfully. The PCDNA3.1-LgE-EGFP-R1gE (Figure 1A) contains the EGFP gene and the LgE and RgE of the gE gene. The PCDNA3.1-LgE-R3gE (Figure 1B) and PCDNA3.1-LgE-R2gE (Figure 1C) only contain the LgE and RgE of the gE gene. The sequences of plasmids were all determined and correct.

### 3.2. Screening of Cell Lines for Bohv-1 Gene Editing

For successful viral gene editing by CRISPR/Cas9, cell lines with high transfection efficiency were chosen using the EGFP-expressing px458 vector. The transfection effect of each cell line is shown in Figure 1D. The transfection efficiency was quantified using flow cytometry (Figure 1E). The results showed that the transfection efficiency was 88.7% in HEK293T cells, 66% in BHK-21 cells, 44.7% in VERO E6 cells, 0.62% in MDBK cells, and 31.3% in BT cells (Figure 1E). Therefore, HEK293T, BHK-21, and VERO E6 cells were found to be candidate cell lines suitable for plasmid transfection. Although HEK293T had high transfection efficiency, BoHV-1 cannot subculture stably in this cell line (Figure 1F). BoHV-1 can be subcultured stably in MDBK, BT, and VERO E6 cells (Figure 1F). Therefore, VERO E6 cells were chosen for subsequent experiments.

### 3.3. sgRNA Editing Efficiency Screening and Identification of Bohv-1 gE/EGFP^+^ Recombinant Virus

Next, we transfected the CRISPR/Cas9 constructs px459-gE-sgRNA1, px459-gE-sgRNA2, px459-gE-sgRNA3, and px459-gE-sgRNA4 into VERO E6 cells and infected them with BoHV-1 12 h later. The virus replication in cells transfected with px459-gE-sgRNA1 and px459-gE-sgRNA2 and their mixture was significantly blocked. Furthermore, the sgRNA-transfected wells showed fewer plaques than the control well (Figure 2A). This indicates that px459-gE-sgRNA1 and px459-gE-sgRNA2 effectively edit the BoHV-1 genome, thereby inhibiting viral replication. We also showed that 2 μg px459-gE-sgRNA1 and px459-gE-sgRNA2 plasmid could effectively inhibit BoHV-1 replication (Figure 2B). Then, px459-gE-sgRNA1, px459-gE-sgRNA2, and PCDNA3.1-LgE-EGFP-R1gE (digested with *HindIII* or *BamHI*) were co-transfected into VERO E6 cells. The sgRNA plasmid and the donor plasmid were co-transfected with 2 μg at a ratio of 0.5:0.5:1. Additionally, BoHV-1 was subsequently inoculated on the cells. The cells were observed at 72 hpi by fluorescence microscopy. The suspected virus was harvested and generated on MDBK cells. The fluorescent cells were thought to be EGFP-positive recombinant viruses (Figure 2C). The fluorescent spot indicated that the recombinant virus was obtained (Figure 2D). The recombinant viruses were purified by six generations of plaque cloning, and the proportion of fluorescent cells in the F6 generation reached more than 95% (Appendix A). We identified the recombinant virus using BoHV-gE F/R primers. The expected product of the parental virus BoHV-1 was 2111 bp and the expected product of the recombinant virus BoHV-1 gE/EGFP^+^ was 1694 bp. (Figure 2E). The BoHV-1 gE/EGFP^+^ recombinant virus was successfully obtained, which provided a preliminary experimental basis for the subsequent knockout of non-essential genes of BoHV-1.

### 3.4. EGFP Was Further Removed from Recombinant BoHV-1 gE/EGFP^+^ by sgRNA of EGFP

Two microgram shear plasmids of px459-EGFP-sgRNA1, px459-EGFP-sgRNA2, and px459-EGFP-sgRNA3 were transfected into VERO E6 cells, which were subsequently infected with BoHV-1 gE/EGFP^+^. We assessed the splicing effect of sgRNA using the plaque assay. The results indicated that the number of viral plaques in cells transfected with px459-EGFP-sgRNA1 and px459-EGFP-sgRNA2 was lower than in the control group (Figure 3A). The shearing efficiency of sgRNA1 and sgRNA2 of EGFP was higher than px459-EGFP-sgRNA3 (Figure 3B). The co-transfection of px459-EGFP-sgRNA1, px459-EGFP-sgRNA2, and PCDNA3.1-LgE-R3gE (digested with *HindIII* or *BamHI*) in VERO E6 cells was performed. Additionally, the treated cells were then infected with BoHV-1 gE/EGFP^+^. The cells with CPE and no fluorescence were suspected to be EGFP-negative recombinant viruses (Figure 3C). The cells were harvested at 72 hpi and generated on MDBK cells. The screened viruses were subjected to six generations of plaque cloning and analyzed by PCR and DNA sequencing. The expected product of recombinant virus BoHV-1 gE/EGFP^−^ amplified with BoHV-gE F/R primers was 587bp, which was smaller than that of the parental virus BoHV-1 (2111bp) and BoHV-1 gE/EGFP^+^ (1694bp) (Figure 3D).

We sequenced the 200 plaques, which we transfected four times. Interestingly, the sequencing results revealed that there were gene insertions, deletions, and even an inversion of the 5′ end and 3′ end of the EGFP sgRNA splicing sites (Figure 4B). In order to test whether the gE gene, the non-coding region between the gE gene and the US9 gene, and the partial US9 gene sequence were deleted simultaneously, px459-EGFP-sgRNA1, px459-EGFP-sgRNA2, and fragments containing the gE genes LgE and RgE (PCDNA3.1-LgE-R2gE donor plasmid was single digested with *HindIII* or *BamHI*) were co-transfected into VERO E6 cells and infected with BoHV-1 gE/EGFP^+^. As described above, four transfections were performed and 50 plaques were screened each time for 200 plaques. Other interesting phenomena appeared when the complete CDS gene sequence containing gE, the partial sequence of US9 and the non-coding region between the gE gene, and the US9 gene were deleted. For example, the recombinant EGFP sequences in the gE CDS region were broken at various locations, and portions of the US4, US6, US7, or US8 sequences were inserted (Figure 4A). In order to verify that this phenomenon was due to the missing part containing the non-coding region between gE and US9 rather than CRISPR/Cas9 off-targets, we randomly selected part of BoHV-1 gene (UL22, UL23, UL27, UL44, UL47, UL53, US4, US6, US7, US8, and US9) for PCR detection and sequencing. The sequencing results showed that there were no mutations in these genes.

### 3.5. Double sgRNAs Editing Increases HR Efficiency

When screening viruses, if the expected recombinant virus appeared, the deletion virus with the simultaneous action of two sgRNAs is often screened. Therefore, we analyzed the editing and homologous recombination efficiency of single sgRNA and double sgRNAs on BoHV-1. The results showed that the linearized HR fragment co-transfected with the two sgRNAs (px459-gE-sgRNA1 and px459-gE-sgRNA2) produced the recombinant virus BoHV-1 gE/EGFP^+^ (Table 2). Two sgRNAs (px459-EGFP-sgRNA1 and px459-EGFP-sgRNA2) also changed BoHV-1 gE/EGFP^+^ to BoHV-1 gE/EGFP^−^, which did not express EGFP. However, no recombinants were screened in the single sgRNA-treated group (Table 3). For double-edited px459-gE-sgRNA1 and px459-gE-sgRNA2, the recombination efficiency was up to 0.5%. Double-edited px459-EGFP-sgRNA1 and px459-EGFP-sgRNA2 mediated recombination efficiency up to 3%. With a similar strategy, we screened another recombinant, BoHV-1 gE/US9/EGFP^−^, which was unsuccessful because the gE, partial US9, and non-coding region between them were deleted. At this time, we found that the recombination efficiency of px459-EGFP-sgRNA1 or px459-EGFP-sgRNA2 was zero, regardless of single or double sgRNA editing. Accordingly, BoHV-1 gE/US9/EGFP^−^ could not be screened out. However, the sgRNA editing efficiency was not significantly different from that of the screened BoHV-1 gE/EGFP^−^ (Table 3 and Table 4). In conclusion, the results suggest that double sgRNAs can increase the chance of homologous recombination. At the same time, we also found that the non-coding region between the gE gene and the US9 gene has a certain effect on the virus. This provides a preliminary experimental basis for the simultaneous deletion of the gE gene and US9 gene in BoHV-1.

### 3.6. Identification of Mutant Viruses In Vitro and Serological Analysis

In order to detect whether the additional US4, gD and gI genes changed the morphology of the virus, we concentrated the virus, stained it with phosphotungstic acid, and observed it under an electron microscope. A mean ± standard deviation analysis showed that the average diameter of BoHV-1 particles is 190 ± 2 nm, while that of BoHV-1 one, BoHV-1 two, BoHV-1 three, BoHV-1 four, and BoHV-1 five particles are 202 ± 2 nm, 185 ± 4.36 nm, 223.4 ± 6.41 nm, 199.2 ± 3.27 nm, and 214.5 ± 1.73 nm, respectively (Figure 5A). The average diameter of the mutant virus was similar to that of the parental virus. To determine whether the BoHV-1 gene-deletion mutants have different replication characteristics, we evaluated the virus proliferation using a one-step growth curve. The results showed that the replication rate and the dynamic trend of virus proliferation in the early stage (within 36 h) of the mutant virus were similar to the parental virus BoHV-1. All viruses grew rapidly over 36 h. The peak titer of BoHV-1 was reached at 48 h. At 48 h, the virus yield of all mutant viruses was significantly different from that of BoHV-1 (*p* < 0.001) (Figure 5B). In addition, at 60 h, BoHV-1 showed 2.25–12.05 times the virus yield compared with all viruses except BoHV-1 three and BoHV-1 four (*p* < 0.001). The virus yield of BoHV-1 was 3.49 times higher than that of BoHV-1 three (*p* < 0.05). There was no difference in the virus yield between BoHV-1 four and BoHV-1. (Figure 5B). To determine the immunogenicity of the mutant virus, we inactivated the virus and immunized BALB/c female mice. To do so, BoHV-1 and DMEM were used as positive and negative controls, respectively. The results showed that parental BoHV-1 induced high levels of neutralizing antibodies after the second immunization. All BoHV-1 mutants also had high neutralizing antibodies titer in mice, which were similar to BoHV-1. More importantly, the neutralization titer of recombinant BoHV-1 three 28d post immunization was significantly different from that of BoHV-1.

### 3.7. Stability of Mutant Viruses

To test the stability of the BoHV-1 mutant, it was subcultured for 20 generations in MDBK cells. BoHV-1 gE/EGFP^+^, BoHV-1 one, BoHV-1 two, BoHV-1 three, BoHV-1 four, and BoHV-1 five could all be stably subcultured, and no back mutation was detected.

## 4. Discussion

Changes and deletions in the envelope protein gE (Us8 homologue) of BoHV-1 do not affect viral replication and immunogenicity but do affect the viral spread between cells and weaken its virulence. Thus, the gE gene is an ideal target gene in the development of gene-deletion BoHV-1. The BoHV-1 gE gene-deletion vaccine is widely used in Europe and North America with good effects [36]. It is difficult to use plasmid based genetic manipulation to edit DNA viruses with large genome sequences, high GC content, and complex functions. Furthermore, CRISPR/Cas9 technology is widely used in viral genome editing [27,35]. In this study, we used CRISPR/Cas9 technology to produce a recombinant BoHV-1 gE/EGFP^+^. We also successfully developed BoHV-1 gE/EGFP^−^ and a series of mutant viruses through reverse screening to remove the EGFP gene. The mutant virus was observed by electron microscopy by phosphotungstic acid staining, and the results showed that the morphology was not affected. In simultaneously immunized mice, the neutralizing antibody titer after 42 days was similar to that of the parent virus, indicating that the immunogenicity was also not affected. The purpose of this study was to establish a CRISPR/Cas9 method that can rapidly knock out the gE gene or gE and US9 genes of BoHV-1 at the same time so that BoHV-1 can become a delivery vector for other viral subunit vaccines. This experiment represents preparations for knocking out multiple genes of BoHV-1 and provides a preliminary basis for the development of genetically engineered subunit vaccines. At the same time, according to the results of this study, we found that the non-coding region between the gE and US9 genes has a certain effect on BoHV-1, which will prevent us from knocking out the non-coding region between the BoHV-1 gene in our subsequent experiments.

Compared with zinc finger nuclease technology and transcription activator-like effector nuclease (TALEN) technology, CRISPR/Cas9 technology is accurate, simple, and efficient. BoHV-1 has a complex genome and high GC content. Traditional methods, such as amplified fragment length polymorphism (AFLP) analysis, T7 endonuclease I (T7EI) detection, and DNA sequencing are not suitable to verify and screen sgRNA. Therefore, in the process of screening effective sgRNAs, we determined the efficiency of sgRNAs by the number of viral plaques. After the virus is inoculated into cells, due to the limitation of low-melting-point agarose, a localized viral plaque is formed. When sgRNA efficiently cleave the viral genome, viral replication in the cells is inhibited, resulting in a reduction in the number of viral plaques (Figure 2A and Figure 3A). In this study, the starting bases of the sgRNA we designed were mainly thymine (T), cytosine (C), or guanine (G), and the CG content was kept as much as possible to about 50%. It has been reported that sgRNAs, beginning with thymine, guanine, or cytosine induce relatively high cleavage efficiency [37]. The sgRNA with a high shearing efficiency can easily screen out mutant viruses. Ineffective sgRNA is similar to using HR technology to produce a recombinant virus, which is very difficult [38]. Konermann S et al. suggested that using two highly active sgRNAs would more effectively and accurately edit the genome, which is one strategy to improve Cas9 specificity and reduce off-target and mutation effects [39]. High GC content stabilizes RNA, which can lead to gRNA/genomic DNA hybridization becoming more stable and more resistant to mismatches [40]. The GC content of BoHV-1 is as high as 70%, which reduces the chance of off-target sgRNA during viral editing [41,42]. Adjusting the sgRNA GC content to 50% or using multiple sgRNA targets simultaneously can improve the shearing efficiency [37]. Therefore, the selection of high-efficiency sgRNA is critical. Only when the target DNA is effectively destroyed by the Cas9 enzyme will the cell initiate DNA repair by HR or NHEJ. Thus, we designed multiple sgRNAs to construct a knockout vector and screened the two sgRNAs with the highest editing efficiency. In the early stage of this study, MDBK cell transfection/infection was used for virus screening, but BoHV-1 gE/EGFP^+^ was not successfully screened. The selection of cell lines with a high transfection efficiency has a great impact on BoHV-1 genome editing. We found that MDBK transfection efficiency was very low. The HEK293T transfection efficiency was 88.7% and the VERO E6 transfection efficiency was 44.7%. BoHV-1 can replicate in VERO E6 cells but cannot replicate in HEK293T cells. Therefore, we chose VERO E6 cells during the process to edit the viral genome. A series of mutant viruses produced by the accurate shear of the Cas9 protein carried by sgRNA were screened. The off-target effect is not random but is related to the target gene and the cell line used in gene editing. Fu et al. showed that off-target mutagenesis produced by different cells is not the same and reducing the Cas9 concentration has no effect on off-target effects [41]. We believe that CRISPR/Cas9 editing positions may change in different cell lines and lead to different HR efficiencies. Further research is needed to explore the differences and related mechanisms.

In this study, the HR efficiency of double sgRNAs was higher than that of single sgRNA. The HR efficiency was as high as 0.5% when using double sgRNAs targeting the BoHV-1 gE gene, namely px459-gE-sgRNA1 and px459-gE-sgRNA2. The HR efficiency was as high as 3.5% when using double sgRNAs targeting the EGFP gene, namely px459-EGFP-sgRNA1 and px459-EGFP-sgRNA2. However, there was little difference in virus replication. Overall, px459-EGFP-sgRNA showed higher HR efficiency than px459-gE-sgRNA. The virulence gene gE plays a pivotal role in the cell-to-cell transmission of BoHV-1 [16]. We speculate that this is because the parental virus has an advantage over the deletion virus in terms of replication and infectivity. The gE deletion virus can only spread outside the cell, which limits the spread of the gE deletion virus. The parental virus can spread directly from cell to cell after infection. In this study, the fluorescence of the F1 recombinant BoHV-1 gE/EGFP^+^ virus was weak. After 36 h, only one fluorescent spot could be observed. After 48 h, the fluorescence range expanded and three fluorescence spots appeared. When using the characteristics of the reporter gene EGFP to reversely screen non-fluorescent plaques, non-fluorescent virus plaques appeared around the fluorescent plaques. We obtained the purified mutant virus after six generations of plaque purification. The proportion of fluorescent cells reached more than 95%, and the virus yield of BoHV-1 gE/EGFP^+^ in the F6 generation was significantly different from that in the F1, F2 and F3 generations (*p* < 0.001) (Appendix A). It showed that the insertion of the EGFP gene into BoHV-1 was successful. We successfully screened BoHV-1 gE/EGFP^−^ by reverse screening px459-EGFP-sgRNA1 and px459-EGFP-sgRNA2. At the same time, the sequence of the reporter gene EGFP located in the gE position of the BoHV-1 virulence gene introduced different types of mutations to viruses without HR. During target-virus screening, we also incidentally detected some mutant viruses. The mutation showed off-target effects, gene insertion, gene deletion, inversion of the 5′ end and 3′ end of the gE EGFP sgRNA splicing site sequence, and large fragment loss (Figure 4A). These phenomena are shown in many CRISPR/Cas9 editing studies. Mutations outside the target range were not found in our study. All screened mutations blocked out the EGFP expression. EGFP fluorescence does not require any substrates or cofactors, and the sequence spontaneously oxidizes to form a cyclized chromophore [43,44] composed of a cycle formed by a serine, tyrosine, and glycine (65–67) tripeptide [45]. By performing sequencing, we found that the loss of Glu located at the 335 site of the EGFP resulted in the loss of the function of EGFP. We also found that the 5′ and 3′ ends of the splicing sites were inverted when two sgRNAs were used simultaneously. We speculate that when the virus is repaired, the sheared fragments are repaired by NHEJ, and the sheared genome was directly inserted into the virus genome. Previous studies reported that the deletion of the US9 gene in BoHV-1 does not affect virus growth or changes [46]. Rijsewijk et al. found that this recombination phenomenon could be observed in NHEJ, mediated by two heterogeneous forms of BoHV-1, which is called the BoHV-1 gE deletion mutant virus. In this mutant, part of the US1.5 gene and part of the US2 gene are duplicated/inverted to repeat at the other end of the US region and appears in the US7 (gI), US8 (gE) and US9 gene positions, while US7 (gI), US8 (gE), and US9 are deleted [47].

In the present study, we used the CRISPR/Cas9 system and the donor plasmid (PCDNA3.1-LgE-R2gE) to attempt to delete the gE gene, the non-coding region between the gE gene and the US9 gene, and the partial US9 gene sequence. The results indicate that the partial sequences of the US4, US6, and US7 genes were inserted into the EGFP sgRNA splicing sites, which deleted the 3′ end of the EGFP sequence. No BoHV-1 gE/US9/EGFP^−^ was produced. We speculate that the non-coding region between the virus gE and US9 may play a role in viral structure and replication. The deletion of the non-coding region led to the weakening of the virus. Therefore, the recombinant virus cannot be screened. Another possibility is that when EGFP sgRNA is used to screen the virus, replication is inhibited and NHEJ occurs, but this phenomenon needs to be further verified. We found that when using CRISPR/Cas9 technology to edit viruses, the efficiency of virus HR is far lower than the efficiency of generating mutant viruses. Liang et al. found that NHEJ is the main method of DSB repair in mammalian cells [34]. NHEJ may occur at any stage of the cell cycle, while HR usually occurs in the late S or G2 stage when the template is available. Furthermore, NHEJ can be completed in approximately 30 min, while HR is significantly slower, taking 7 h or longer to complete. Therefore, the efficiency of viral HR is much lower than the efficiency of producing mutant viruses.

We demonstrate that efficient CRISPR/Cas9-mediated HR can quickly transform the BoHV-1 virus. When using double sgRNAs to edit live viruses, the HR efficiency is much higher than when using single sgRNA. Indeed, we achieved a recombination efficiency of up to 3%, which is the highest recombination efficiency produced in live BoHV-1 editing. We developed four high-efficiency sgRNAs. Among them, when EGFP sgRNA is used to screen BoHV-1 deletion viruses, the highest single sgRNA editing efficiency is up to 5.5%. However, it is easier and more efficient to use double sgRNAs to screen viruses than single sgRNA, which allows for an editing efficiency of up to 9%.

## 5. Conclusions

In summary, we developed a series of recombinant gE-deletion BoHV-1 viruses using the CRISPR/Cas9 gene editing system, which recombined EGFP^+^ and EGFP^−^. We discovered some interesting phenomena. The above two points lay the foundation for future research on BoHV-1. In addition, the CRISPR/Cas9 system was combined with a homologous recombination to analyze the recombination efficiency and we screened four pairs of sgRNA sequences with high efficient splicing, which proved that the CRISPR/Cas9 system is a powerful tool for screening sgRNAs and rapidly generating mutant viruses. We also found that the non-coding region between the gE and US9 genes has a certain effect on BoHV-1, which will prevent us from knocking out this non-coding of BoHV-1 gene in our subsequent experiments.

## Figures and Tables

**Figure 1 vetsci-09-00166-f001:**
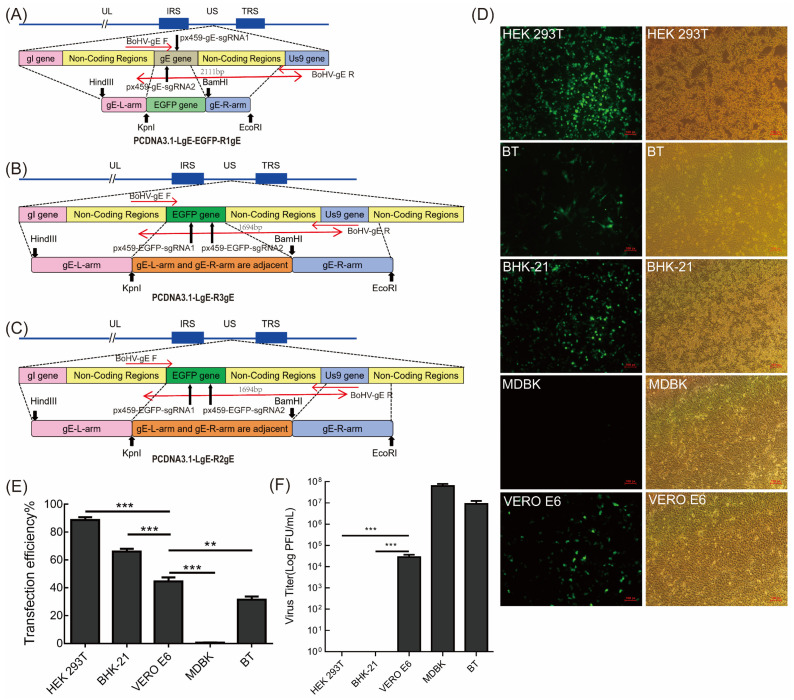
Construction of the BoHV-1 ΔgE donor plasmid and determination of the cell lines for transfection. (**A**–**C**) Schematic showing the donor plasmid sequence used to edit BoHV-1 and the re-combination sites inserted into the BoHV-1 genome. The long black arrows represent the sgRNA cleavage site, the short black arrows represent the restriction enzyme cutting site, and the short red arrows and long red arrows represent the primer position and product size, respectively. (**D**) The transfection efficiency of the empty px458 vector in different cell lines. Fluorescence was evaluated after 48 h. (**E**) Flow cytometry analysis of HEK293T, BHK-21, VERO E6, MDBK, and BT cell transfection efficiency. (**F**) BoHV-1 virus titer in different cells. The 80% confluent MDBK, BT, HEK293T, and VERO E6 cells were infected with 10-fold proportional serial dilutions of the virus. There were significant differences between the different cell lines. The data are shown as mean ± SD; ** *p* < 0.01, and *** *p* < 0.001.

**Figure 2 vetsci-09-00166-f002:**
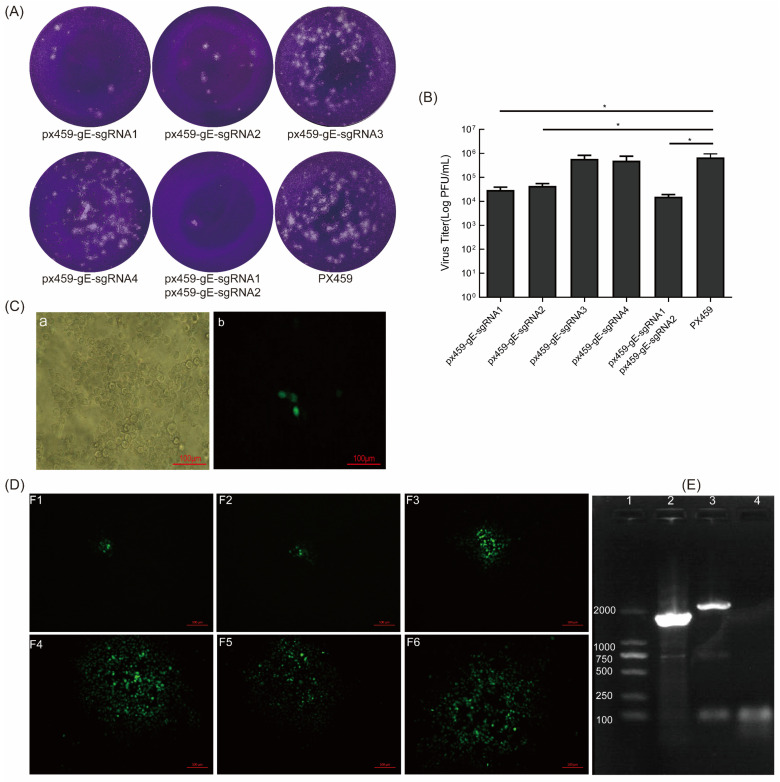
sgRNA of gE inhibits BoHV-1 replication and screening of BoHV-1 gE/EGFP^+^. (**A**) sgRNA inhibits the formation of BoHV-1 plaques. A total of 2 μg of px459-gE-sgRNA1, 2, 3, 4, and px459 were co-transfected into VERO E6 cells. After 12 h, the cells were infected with BoHV-1 (MOI = 1). The virus was collected at 72 hpi and a plaque assay was performed. (**B**) The virus titer of BoHV-1 when it was proceeded with sgRNA of gE on MDBK cells. (**C**) The fluorescent cells indicated that BoHV-1 gE/EGFP^+^ was produced. (**Ca**) bright field; (**Cb**) F0 of rescued recombinant virus. (**D**) The fluorescent spot when cloning and purifying the EGFP-positive viruses. (**E**) PCR identification of BoHV-1 gE/EGFP^+^. The viral genome was extracted from the supernatant after the purification of sixth-generation clones. PCR amplification was performed using BoHV-gE-F/R primers. The negative control was double distilled water. Lane 1: DNA marker 2000, lane 2: BoHV-1gE/EGFP^+^, lane 3: wtBoHV-1, lane 4: negative control. The data are shown as mean ± SD; * *p* < 0.05.

**Figure 3 vetsci-09-00166-f003:**
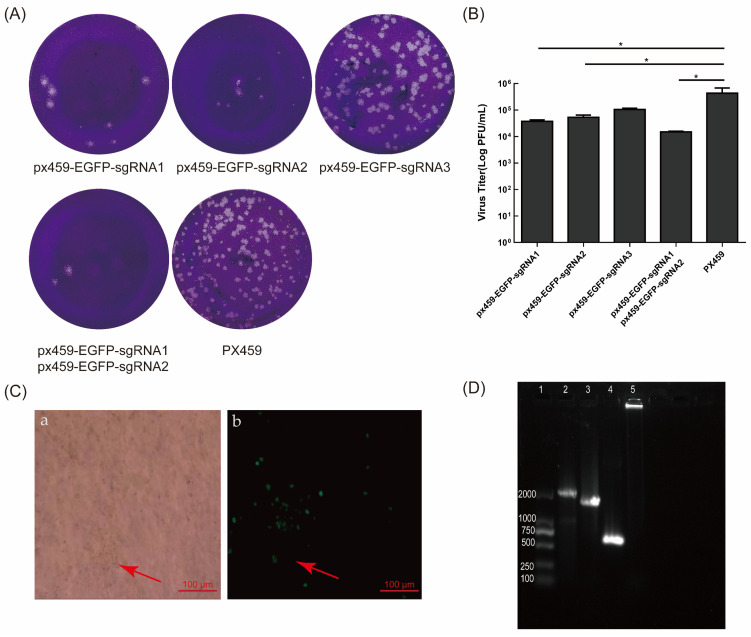
sgRNA of EGFP inhibited the formation of BoHV-1 gE/EGFP^+^ plaques and the targeted deletion of EGFP. (**A**) The plaques with different sgRNAs of EGFP. Two micrograms of px459-gE-sgRNA1, 2, 3, or px459 were transfected into VERO E6 cells. After 12 h, the cells were infected with BoHV-1 gE/EGFP^+^ (MOI = 1) and the virus was fixed and stained for plaque testing at 72 hpi. (**B**) The virus titer with sgRNA of EGFP editing on MDBK cells. (**C**) The cytopathic effect and no fluorescence of BoHV-1 gE/EGFP^−^ (**Ca**) bright field; (**Cb**) fluorescence field. The red arrow indicates first-generation BoHV-1 gE/EGFP^−^ recombinant viruses. (**D**) PCR identification of BoHV-1 gE/EGFP^−^. The purified viral genome was extracted and PCR amplification was performed using BHV-gE-F/R primers. The negative control was double distilled water. Lane 1: DNA marker 2000, lane 2: wtBoHV-1, lane 3: BoHV-1gE/EGFP^+^, lane 4: BoHV-1gE/EGFP^−^, lane 5: negative control. The data are shown as mean ± SD; * *p* < 0.05.

**Figure 4 vetsci-09-00166-f004:**
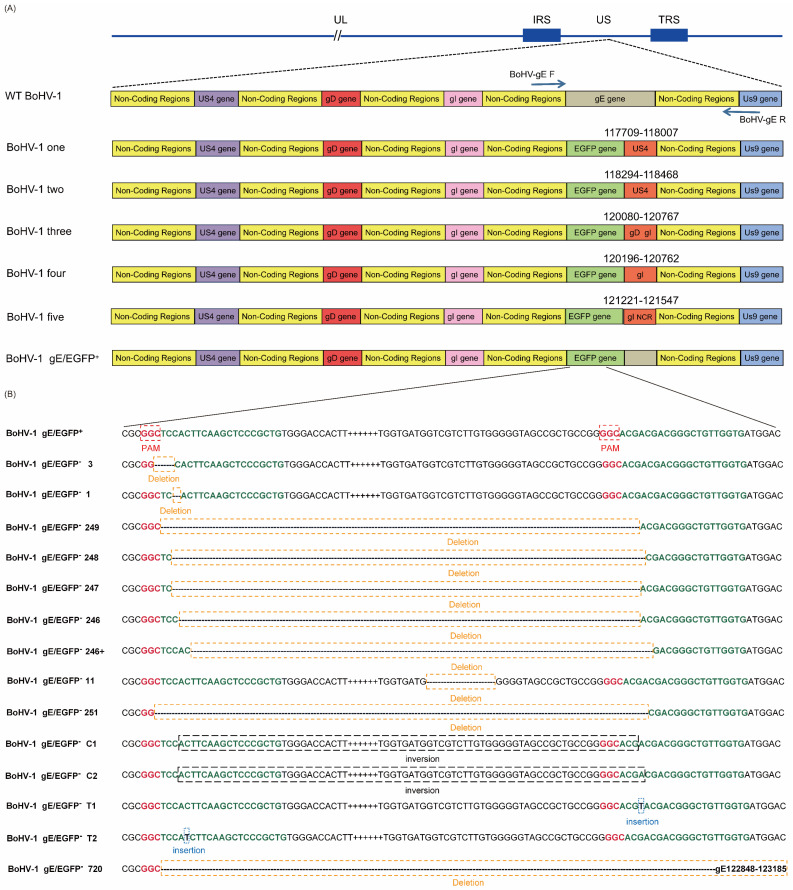
Editing CRISPR/Cas9 induces high-efficiency mutations in the BoHV-1 genome. (**A**) Summary of representative mutations of BoHV-1 induced with px459-EGFP-sgRNA1 and px459-EGFP-sgRNA2 appearing outside the EGFP sequence. (**B**) Summary of representative mutations of BoHV-1 induced with px459-EGFP-sgRNA1 and px459-EGFP-sgRNA2 appearing inside the EGFP sequence.

**Figure 5 vetsci-09-00166-f005:**
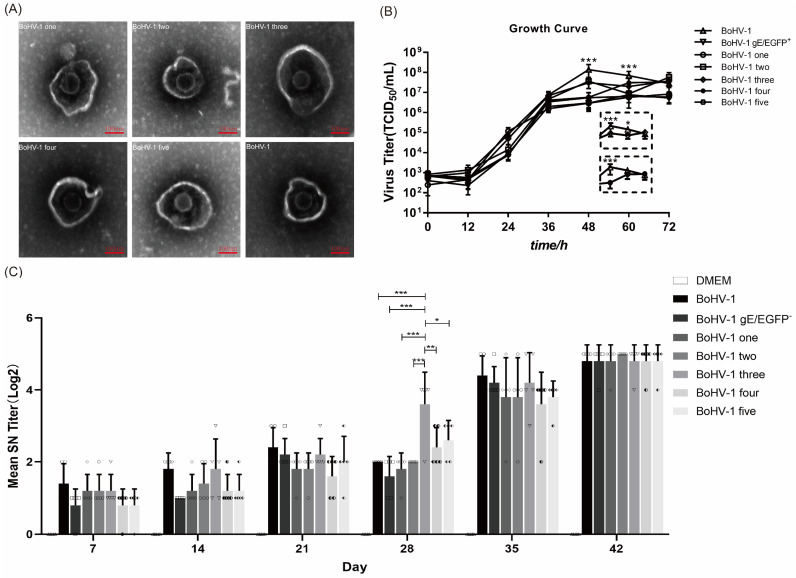
The insertion of genes did not change the replication, morphology or immunogenicity of the virus. (**A**) After staining with phosphotungstic acid, the morphology of BoHV-1 was compared with that of other mutant viruses using an electron microscope. (**B**) BoHV-1, BoHV-1 gE/EGFP^−^, and other recombinant viruses inoculate MDBK cells at 0.01 MOI. At 12, 24, 36, 48, 60, and 72 hpi after inoculation, the Karber method was used to calculate the virus titer and draw the virus growth curve. Windows shows the respective titers of parental BoHV-1, mutant three and four at 48, 60 and 72 h post infection. (**C**) The virus was injected intramuscularly, and the second immunization was performed at 21 d after the first immunization. Serum was collected every 7 days after the initial immunization to detect specific neutralizing antibodies against BoHV-1. *** *p* < 0.001, ** *p* < 0.01, * *p* < 0.05. Each graph represents the neutralizing antibody titer of each mouse in different immunization groups. Circle: BoHV-1; Square: BoHV-1 gE/EGFP^−^; Diamond: BoHV-1 one; Hexagon: BoHV-1 two; Inverted triangle: BoHV-1 three; Left half-solid hexagon: BoHV-1 four; left half filled diamond: BoHV-1 five.

**Table 1 vetsci-09-00166-t001:** Sequences and positions of the primers and sgRNAs utilized in this study.

Primers	Nucleotide Sequences (5′-3′)	Genome Position	Restriction Sites
LgE-F	aagcttTGCTCTTCTCCATCGCCCATC	120663–120683	*HindIII*
LgE-R	ggtaccCATTGCCAAATGCCCTTTTCGA	121695–121716	*KpnI*
EGFP-F	ggtaccATGGTGAGCAAGGGCGA	1–17	*KpnI*
EGFP-R	ggatccCTTGTACAGCTCGTCC	702–717	*BamHI*
R1gE-F	ggatccAGTCGTTACTTCGGACCGTTTGGTGC	122847–122872	*BamHI*
R1gE-R	gaattcTCAGCGCCTCGATAGTTTTCGTTGAC	123569–123594	*EcoRI*
R2gE-F	ggatccTCACCATCGAGGACGCGCCGGCCAGCGCAGA	123663–123693	*BamHI*
R2gE-R	gaattcCGAATCCTCGGCCGGCCCGAATCCCCTCCTT	124332–124362	*EcoRI*
R3gE-F	ggatccCTCAAGTCCATCCTCCGCTAG	123421–123441	*BamHI*
R3gE-R	gaattcGCCCTTGTCATATTTTTTTAA	124486–124506	*EcoRI*
BoHV-gE F	CGCCGGGTTGTTAAATGGGTCTCG	121573–121596	
BoHV-gE R	CGGGCGCGTCCTCGATGGTG	123664–123683	
px459-EGFP-sgRNA1 F	caccgGTCGCCCTCGAACTTCACCT	335–354	*BbsI*
px459-EGFP-sgRNA1 R	aaacAGGTGAAGTTCGAGGGCGACc	335–354	*BbsI*
px459-EGFP-sgRNA2 F	caccgGTGGTTGTCGGGCAGCAGCA	581–600	*BbsI*
px459-EGFP-sgRNA2 R	aaacTGCTGCTGCCCGACAACCACCc	581–600	*BbsI*
px459-EGFP-sgRNA3 F	caccgGTTGGGGTCTTTGCTCAGGG	620–639	*BbsI*
px459-EGFP-sgRNA3 R	aaacCCCTGAGCAAAGACCCCAACc	620–639	*BbsI*
px459-gE-sgRNA1 F	caccgCGGCGACGAGGAGACGCAGTTGG	122217–122239	*BbsI*
px459-gE-sgRNA1 R	aaacCCAACTGCGTCTCCTCGTCGCCGc	122217–122239	*BbsI*
px459-gE-sgRNA2 F	caccgCGCCGATGAGCCGGTCGTACAGG	122190–122212	*BbsI*
px459-gE-sgRNA2 R	aaacCCTGTACGACCGGCTCATCGGCGc	122190–122212	*BbsI*
px459-gE-sgRNA3 F	caccgCGAGCCCGGGGTTTCGGTCGCGG	121802–121824	*BbsI*
px459-gE-sgRNA3 R	aaacCCGCGACCGAAACCCCGGGCTCGc	121802–121824	*BbsI*
px459-gE-sgRNA4 F	caccgCCACGTCGGTGAAGCACTCGCGG	121977–121999	*BbsI*
px459-gE-sgRNA4 R	aaacCCGCGAGTGCTTCACCGACGTGGc	121977–121999	*BbsI*
UL53 F	CACTGAGACCGGCATTTTA	2971–2989	
UL53 R	CGAAGAGTTTATTGCTGAC	4376–4394	
UL47 F	ACTTGGGTCTACACGGGATTTA	11902–11923	
UL47 R	TTGTCCTGCTTGTGCTTGAACG	14783–14804	
UL44 F	GACGACTACGAAAACTAC	16459–16476	
UL44 R	GACCACGAAAGCACAAAA	18318–18335	
UL27 F	CAGTTTTTTTGCTTCGCATCCG	55179–55200	
UL27 R	TTTTGCATTACTTTTGGGGTCA	58536–58557	
UL23 F	AAAAACGGCACGTCTTCAGCTC	62955–62976	
UL23 R	ACCACCATTTCCCACTCTTCGA	64582–64603	
UL22 F	GACCCCAGTTGTGATGAATGCA	63995–64016	
UL22 R	GCCGTCGGACAGTGAGTATGAG	67196–67217	
US4 F	CAGATGCTGACCTTTGACTTTC	117119–117140	
US4 R	GTTTAACTCGCAATAGACACGC	118755–118776	
US6 F	GACGCAGCGGTGGTGGTGATGT	118381–118402	
US6 R	GCGGATGGGCGATGGAGAAGAG	120665–120686	
US7 F	TGCCCAGAAAGCCAAAAAAG	120085–120104	
US7 R	TCAGCCCGAAAAGCAATAAC	121767–121786	
US8 F	CGCGAGAGGGTTCGAAAAGGGC	121684–121705	
US8 R	CGCCTCGATAGTTTTCGTTGAC	123569–123590	
US9 F	CTGTGCCGTCTGACGGAAAGCA	123461–123482	
US9 R	TATATCTTGTGGTTCTAGTTGTT	124429–124451	

**Table 2 vetsci-09-00166-t002:** The gene editing in BoHV-1 gE/EGFP^+^ using px459-gE-sgRNA1/px459-gE-sgRNA2/px459.

	px459-gE-sgRNA1/px459	px459-gE-sgRNA2/px459	px459-gE-sgRNA1/sgRNA2
BoHV-1 gE/EGFP^+^	0	0	1
Plaque number	200	200	200
HR efficiency	0	0	0.5%

Summary of the phenomena that occur in BoHV-1 gE/EGFP^+^ using px459-gE-sgRNA1/px459-gE-sgRNA2/px459. The above experiment was divided into three groups to screen recombinant virus BoHV-1 gE/EGFP^+^. The first group px459-gE-sgRNA1/PX459: the shear plasmids PX459-gE-sgRNA1 and px459 and the donor plasmid PCDNA3.1-LgE-EGFP-R1gE were co-transfected and infected with BoHV-1. The second group px459-gE-sgRNA2/PX459: the shear plasmids PX459-gE-sgRNA2 and px459 and the donor plasmid PCDNA3.1-LgE-EGFP-R1gE were co-transfected and infected with BoHV-1. The third group px459-gE-sgRNA1/PX459-gE-sgRNA2: the shear plasmids PX459-gE-sgRNA1 and PX459-gE-sgRNA2 and the donor plasmid PCDNA3.1-LgE-EGFP-R1gE were co-transfected and infected with BoHV-1. The above three groups were transfected four times. The dosage of plasmid per transfection in each group was 2 μg and the ratio was 0.5:0.5:1. Fifty plaques were screened each time (a total of 200 plaques were screened) and the recombination efficiency was analyzed.

**Table 3 vetsci-09-00166-t003:** The gene editing in BoHV-1 gE/EGFP^−^ using px459-EGFP-sgRNA1/px459-EGFP-sgRNA2/px459.

	px459-EGFP-sgRNA1/px459	px459-EGFP-sgRNA2/px459	px459-EGFP-sgRNA1/sgRNA2
Edited EGFP	7	3	16
BoHV-1 gE/EGFP^−^	0	0	6
Plaque number	200	200	200
sgRNA editing efficiency	3.5%	1.5%	8%
HR efficiency	0	0	3%

Summary of the phenomena that occur in BoHV-1 gE/EGFP^−^ using px459-EGFP-sgRNA1/px459-EGFP-sgRNA2/px459. The above experiment was divided into three groups to screen recombinant virus BoHV-1 gE/EGFP^−^. The first group px459-EGFP-sgRNA1/PX459: the shear plasmids px459-EGFP-sgRNA1 and px459 and the donor plasmid PCDNA3.1-LgE-EGFP-R3gE were co-transfected and infected with BoHV-1 gE/EGFP^+^. The second group px459-EGFP-sgRNA2/PX459: the shear plasmids px459-EGFP-sgRNA1 and px459 and the donor plasmid PCDNA3.1-LgE-EGFP-R3gE were co-transfected and infected with BoHV-1 gE/EGFP^+^. The third group px459-EGFP-sgRNA1/px459-EGFP-sgRNA2: the shear plasmids PX459-gE-sgRNA1 and PX459-gE-sgRNA2 and the donor plasmid PCDNA3.1-LgE-EGFP-R3gE were co-transfected and infected with BoHV-1 gE/EGFP^+^. The above three groups were transfected four times. The dosage of plasmid per transfection in each group was 2 μg and the ratio was 0.5:0.5:1. Fifty plaques were screened each time (a total of 200 plaques were screened) and the recombination efficiency was analyzed.

**Table 4 vetsci-09-00166-t004:** Summary of the phenomena that occur in BoHV-1 gE /US9/EGFP^−^ using px459-EGFP-sgRNA1/px459-EGFP-sgRNA2/px459.

	px459-EGFP-sgRNA1/px459	px459-EGFP-sgRNA2/px459	px459-EGFP-sgRNA1/sgRNA2
Edited EGFP	11	4	18
BoHV-1 gE/US9/EGFP^−^	0	0	0
Plaque number	200	200	200
sgRNA editing efficiency	5.5%	2%	9%
HR efficiency	0	0	0

Summary of the phenomena that occur in BoHV-1 gE /US9/EGFP^−^ using px459-EGFP-sgRNA1/px459-EGFP-sgRNA2/px459. The above experiment was divided into three groups to screen recombinant virus BoHV-1 gE/US9/EGFP^−^. The first group px459-EGFP-sgRNA1/PX459: the shear plasmids px459-EGFP-sgRNA1 and px459 and the donor plasmid PCDNA3.1-LgE-EGFP-R2gE were co-transfected and infected with BoHV-1 gE/EGFP^+^. The second group px459-EGFP-sgRNA2/PX459: the shear plasmids px459-EGFP-sgRNA1 and px459 and the donor plasmid PCDNA3.1-LgE-EGFP-R2gE were co-transfected and infected with BoHV-1 gE/EGFP^+^. The third group px459-EGFP-sgRNA1/px459-EGFP-sgRNA2: the shear plasmids PX459-gE-sgRNA1 and PX459-gE-sgRNA2 and the donor plasmid PCDNA3.1-LgE-EGFP-R2gE were co-transfected and infected with BoHV-1 gE/EGFP^+^. The above three groups were transfected four times. The dosage of plasmid per transfection in each group was 2 μg and the ratio was 0.5:0.5:1. Fifty plaques were screened each time (a total of 200 plaques were screened) and the recombination efficiency was analyzed.

## Data Availability

The data presented in this study are available on request from the corresponding author.

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
