# Peer review of "Concurrent Gene Insertion, Deletion, and Inversion during the Construction of a Novel Attenuated BoHV-1 Using CRISPR/Cas9 Genome Editing"

_vetsci, 2022, doi:10.3390/vetsci9040166_

Round 1
Reviewer 1 Report
Liu and colleagues describe here the generation of a glycoprotein (g)E deleted bovine alpha herpesvirus 1 (BoHV-1) using CRISPR/Cas9 gene editing combined with plasmids for homologous repair with the aim to build “a foundation for the research of BoHV-1 and vaccine development in the future (abstract lines 21/22)”.
BoHV-1 is still a problem in cattle and final eradication is difficult partly due to latency as it is for other animal herpesviruses. Attenuated gene-deleted marker vaccines have been shown to be extremely useful and the success in the fight against Aujeszky’s Disease in many countries is mainly based on gE-deleted pseudorabies vaccine viruses which served as a blueprint for the control of other diseases caused by (herpes)virus.
Although the search for straightforward ways for gene manipulation and vaccination strategies is still relevant the paper unfortunately suffers from a lack of original information. Successful and easy mutant generation via CRISPR/Cas9 is repeatedly described and is nowadays widely used in many labs and for a vast number of different (herpes) viruses. gE-deleted BoHV-1 mutants have been generated before. Although it is of interest for the authors lab to show how efficient the mutagenesis protocol is, the relevance for other researchers in the field is limited.
The manuscript also contains a number of misleading or even incorrect statements. Some examples are given below:
Lines 34/35: “The latent infection of BoHV-1 makes disease control more difficult. Therefore, it is necessary to develop new vaccines that can distinguish wild virus infection from vaccine immunity.” The new vaccine suggested here will not help to solve the problem with latency of herpesviruses and gE-deleted BoHV-1 is already used in the DIVA strategy.
Lines 390/391: “This is similar to the results of previous studies that reported that the insertion of the viral genome increased the diameter of the virus particle”. In the studies cited a bullet-shaped Rhabdovirus (10K genome) was investigated. If the incorporation of a slightly increased BoHV-1 genome (135K) due to gene duplications would increase particle size the capsid should be enlarged, but this was not investigated here and was never observed before despite numerous mutants were published comprising even larger additional DNA.
Lines 423/424: “It is difficult to genetically manipulate DNA viruses with large genome sequences, high GC content, and complex functions”. There is a huge number of BoHV-1 mutants described (see papers from labs of e.g. Bieńkowska-Szewczyk, Chowdhury, Jones, van Drunen Littel-van den Hurk and many others) and also for PRV which has an even higher GC content!
Beside the mentioned passages, the manuscript requires thorough corrections not only for English language and grammar but even more for precise and correct scientific wording and content. It is difficult to follow the experimental strategy and the outcome.
Author Response
Response to Reviewer 1 Comments
Point 1: Lines 34/35: “The latent infection of BoHV-1 makes disease control more difficult. Therefore, it is necessary to develop new vaccines that can distinguish wild virus infection from vaccine immunity.” The new vaccine suggested here will not help to solve the problem with latency of herpesviruses and gE-deleted BoHV-1 is already used in the DIVA strategy.
Response 1: We apologize for the lack of detail stated in the manuscript. The purpose of this study is to establish a CRISPR/Cas9 method that can rapidly knock out the gE gene or gE and US9 genes of BoHV-1 at the same time, so that BoHV-1 can become a delivery vector for other viral subunit vaccines. In this study, we developed a series of recombinant gE-deletion BoHV-1 by CRISPR/Cas9 gene editing system, and these mutant viruses had neutralizing antibody titer resemblance to the parental virus. And related studies have reported that the BoHV-1 gE gene will not be detoxified through the nasal cavity even if it is reactivated after deletion [1,2]. Deletion of the gE gene reduces viral neurotropism, thereby reducing latency and risk of reactivation.
At present, research on the prevention and treatment of latent human herpesvirus infection mainly focuses on two points: one is vaccination before primary infection, and the other is the early application of antiviral drugs. Relevant studies have shown that 90% of people have latent infection with herpes virus, and the peak age of forming this infection is in early childhood. Therefore, early immunization to prevent latent infection of herpes virus is considered to be the most likely effective method to prevent latent infection of trigeminal ganglion.
Unfortunately, our study is not enough to address the issue of the incubation period of herpes virus. We suspect that, based on the idea that vaccination is one of the ways to avoid latent infection, this problem can be avoided if inactivated vaccines are used.
The safety of live attenuated vaccines is an important issue that deserves our attention. Live-attenuated vaccines allow the virus to establish a life-long incubation period that, once reactivated by external environmental stimuli, spreads in cattle. Furthermore, most live attenuated vaccines cause immunosuppression in calves whose immune systems are not fully developed, resulting in disease. And when the vaccine strain is replicated in the same bovine, there is a risk of strong virulence.
This study lays the foundation for the subsequent knockout of multiple genes in BoHV-1 and making BoHV-1 a delivery vector. Meanwhile, after editing the BoHV-1 genome using the CRISPR/Cas9 system, we discovered some interesting phenomena. The above two points lay the foundation for future research on BoHV-1. In addition, the CRISPR/Cas9 system was combined with homologous recombination to analyze the recombination efficiency and screened 4 pairs of sgRNA sequences with high efficient splicing, which proved that the CRISPR/Cas9 system is a powerful tool for screening sgRNAs and rapidly generating mutant viruses. we also found that the non-coding region between the gE and US9 genes has a certain effect on BoHV-1, which can prevent us from knocking out the non-coding region between the BoHV-1 gene in our subsequent experiments.
- Brum MC, Coats C, Sangena RB, Doster A, Jones C, Chowdhury SI. Bovine herpesvirus type 1 (BoHV-1) anterograde neuronal transport from trigeminal ganglia to nose and eye requires glycoprotein E. J Neurovirol. 2009 Apr;15(2):196-201. doi: 10.1080/13550280802549605. PMID: 19115127.
- Earley B, Tiernan K, Duffy C, Dunn A, Waters S, Morrison S, McGee M. Effect of suckler cow vaccination against glycoprotein E (gE)-negative bovine herpesvirus type 1 (BoHV-1) on passive immunity and physiological response to subsequent bovine respiratory disease vaccination of their progeny. Res Vet Sci. 2018 Jun;118:43-51. doi: 10.1016/j.rvsc.2018.01.005. Epub 2018 Jan 10. PMID: 29421483; PMCID: PMC7111865.
Point 2: Lines 390/391: “This is similar to the results of previous studies that reported that the insertion of the viral genome increased the diameter of the virus particle”. In the studies cited a bullet-shaped Rhabdovirus (10K genome) was investigated. If the incorporation of a slightly increased BoHV-1 genome (135K) due to gene duplications would increase particle size the capsid should be enlarged, but this was not investigated here and was never observed before despite numerous mutants were published comprising even larger additional DNA.
Response 2: Thanks for your question. The particle diameter of mature BoHV-1 is approximately 150-200 nm. In this study, a series of mutant viruses were generated by simultaneously knocking out the gE and US9 genes of BoHV-1 and the noncoding regions between them. Comparing the morphological structure of the mutant virus and the parental virus by electron microscope observation, we found that the particle diameter of BoHV-1 three and BoHV-1 five has changed, but the shape has not changed. Through data review, we found that the sizes of rabies viruses expressing 2 G proteins and 3 G proteins were different[1], so the same conclusion was reached based on the results of this experiment, but the specific reasons for the change in virus diameter were not studied in detail. We speculate that this may be due to the insertion of other sequences into the viral genome or the action of heavy metal salts. Based on the questions raised by the reviewer, we have performed a statistical analysis and found that the diameters of the viruses are not significantly different, and have revised the corresponding paragraphs in the manuscript(We marked it in red on page 13, lines 415-419).The reviewer's attitude towards scientific research is very worth learning.
- Li C, Zhang H, Ji L, Wang X, Wen Y, Li G, Fu ZF, Yang Y. Deficient Incorporation of Rabies Virus Glycoprotein into Virions Enhances Virus-Induced Immune Evasion and Viral Pathogenicity. Viruses. 2019 Mar 4;11(3):218. doi: 10.3390/v11030218. PMID: 30836694; PMCID: PMC6466124.
Point 3: Lines 423/424: “It is difficult to genetically manipulate DNA viruses with large genome sequences, high GC content, and complex functions”. There is a huge number of BoHV-1 mutants described (see papers from labs of e.g. Bieńkowska-Szewczyk, Chowdhury, Jones, van Drunen Littel-van den Hurk and many others) and also for PRV which has an even higher GC content!
Response 3: Thanks to the reviewer for the suggestion. It has been reported that sgRNAs starting from thymine, guanine or cytosine induce relatively high cleavage efficiency. Furthermore, sgRNAs with high AT content induce higher cleavage efficiency than sgRNAs with high GC content, which may be due to the fact that more energy is required to unravel the three hydrogen bonds between guanine and cytosine in the DNA double helix structure[2]. Therefore, this sentence is due to the consideration of the overall size of the BoHV-1 genome and the GC content, which has been explained in the discussion section according to the reviewer's suggestion(We marked in red on page 14, lines 456-458 and on page 15, lines 488-495 and 500-503).
- Zeng B, Zhan S, Wang Y, Huang Y, Xu J, Liu Q, Li Z, Huang Y, Tan A. Expansion of CRISPR targeting sites in Bombyx mori. Insect Biochem Mol Biol. 2016 May;72:31-40. doi: 10.1016/j.ibmb.2016.03.006. Epub 2016 Mar 24. PMID: 27032928.
Point 4: Beside the mentioned passages, the manuscript requires thorough corrections not only for English language and grammar but even more for precise and correct scientific wording and content. It is difficult to follow the experimental strategy and the outcome.
Response 4: We apologize for the inappropriate use of scientific wording. We have made precise revisions to the manuscript as requested by the reviewer and reworked language polish. Thanks for your suggestion(We marked lines 88-90, 98-100,123-125,148-149,155-156,164-166,173-177,182-183,198-200,221,225-230,239-240,269-270,277-279,293-296,303-305,323-325,330-332,347-348,350-352,364-368,379-381,394-395,409-410,415-419,421-428,456-458,465-473,479-495,500-503,533-537,551-553,563-567,571-573,575 and 589-600 in red font).

Reviewer 2 Report
The authors targeted the gE gene of BoHV-1 virus and provided an available method of virus knockout by CRISPR/Cas9 system. I consider this study can be published but needs major revision. There are many grammar and spelling mistakes in this article.
In method:
- If the dosage of all the transfected sgRNA is 2ug? If not, please clarify clearly in each part.
- Why did author use PX458 plasmid for transfection efficiency analysis, but performed experiments with PX459? Why did authors use PX458 plasmid in the whole study? The authors used PX459 for gene editing, please describe whether add puromycin for selection.
In results:
- In figure 1A, please mark the site of sgRNAs. And the base pair in figure 1C is incorrect.
- In 3.4, why did removing EGFP affect the number of viral plaques in figure 3A, please clarify in Discussion part.
- In 3.6, the diameter of BoHV-1 particles needs to be performed statistical analysis.
- The authors described mutant virus replicate slower than the parent virus, but the data seems not to be significantly different.
- The authors think that the more important is “the neutralization titer of recombinant BoHV-1 three post immunization 28 d is significantly different from that of BoHV-1”. Please explain why.
In discussion:
- The authors discussed the effect of sgRNA GC content on editing efficiency, but authors should illustrate the GC content of the sgRNA you designed and discuss how it affect efficiency in the present study.
- At page 15 line 481, the authors mentioned that the proportion of fluorescent cells, please provide the proportion of each generation as supplementary materials.
- At page 16 line 515, NHEJ is the main method of DSB repair, not HR.
References
- Please cite some latest references
Author Response
Response to Reviewer 2 Comments
Point 1: If the dosage of all the transfected sgRNA is 2ug? If not, please clarify clearly in each part.
Response 1: The reviewer's attitude towards scientific research is very worth learning. We are sorry for not describing in detail the dosage of sgRNA used. In this study, the total dosage of sgRNA transfected in each group was 2μg. If the sgRNA plasmid and the donor plasmid were co-transfected, their ratio was 0.5:0.5:1, and the total dosage was also 2μg. Based on the reviewer's suggestions, revisions have been made in the corresponding locations of the manuscript (We marked lines 164-166, 269-270, 293-294, 379-381, 394-395 and 409-410 in red font).
Point 2: Why did author use PX458 plasmid for transfection efficiency analysis, but performed experiments with PX459? Why did authors use PX458 plasmid in the whole study? The authors used PX459 for gene editing, please describe whether add puromycin for selection.
Response 2: PX458 was used because the EGFP fluorescent gene carried by it can be used for flow cytometric sorting, and the transfection efficiency of different cell lines can be determined by the number of fluorescent cells. When screening the recombinant virus, the EGFP fluorescent signal in the infected cells after inoculation is the criterion for judging the success of the recombination, and the EGFP fluorescent gene carried by the PX458 plasmid will cause false positive results. Therefore, the recombinant virus screening was performed using the PX459 plasmid that does not carry the EGFP fluorescent gene. When using PX459 to edit the viral genome, we did not add puromycin for screening, because we screened the recombinant virus by the EGFP fluorescence signal of infected cells after inoculation.
Point 3: In figure 1A, please mark the site of sgRNAs. And the base pair in figure 1C is incorrect.
Response 3: Following the reviewer's suggestion, we have added sgRNA cleavage positions to Figure 1A, 1B, and 1C. We are sorry for the error in Figure 1C, our understanding is that the BoHV-1 gE/EGFP+ recombinant virus was detected using the BoHV-gE F/R primer, and its PCR product was 1694 bp. What we are labeling is genome length, not post-recombination length.
Point 4: In 3.4, why did removing EGFP affect the number of viral plaques in figure 3A, please clarify in Discussion part.
Response 4: After the virus is inoculated into cells, due to the limitation of low melting point agarose, it can only replicate in the initially infected cells, and after a replication cycle, the progeny virus is released and expanded to the surrounding area. After several cycles of proliferation, a localized viral plaque is formed. Theoretically, a plaque is formed by the replication of a virus particle that originally infected the cell. In the process of screening effective sgRNAs, we determined the efficiency of sgRNAs by the number of viral plaques. Due to the complex genome and high GC content of BoHV-1, traditional methods such as amplified fragment length polymorphism analysis (AFLP), T7 endonuclease I detection (T7EI), and DNA sequencing are not suitable for validating and screening sgRNAs. When px459-gE-sgRNA1/px459-gE-sgRNA2 and px459-EGFP-sgRNA1/px459-EGFP-sgRNA2 efficiently cleave the viral genome, viral replication in cells is inhibited, resulting in a reduction in the number of viral plaques.
Added to the Discussion section at the reviewer's suggestion (indicated in red on page 15, lines 479-495).
Point 5: In 3.6, the diameter of BoHV-1 particles needs to be performed statistical analysis.
Response 5: Thanks to the reviewer for the suggestion. We have performed a statistical analysis as you requested and found that the diameters of virus particles are not significantly different and have been revised in the manuscript(We marked it in red on page 13, lines 415-419).
Point 6: The authors described mutant virus replicate slower than the parent virus, but the data seems not to be significantly different.
Response 6: We are sorry for not marking the significance in detail. According to the reviewer's comments, figure 5 B is marked, and the problem is explained in the results(We marked it in red on page 13, lines 421-428).
Point 7: The authors think that the more important is “the neutralization titer of recombinant BoHV-1 three post immunization 28 d is significantly different from that of BoHV-1”. Please explain why.
Response 7: Thanks for your question. Through discussion, we speculate that there are three possibilities: 1. Due to the recombination of part of the gD and gI gene sequences into the virus genome, because the gD protein can induce neutralizing antibodies against BoHV-1 infection. 2. Due to different age, weight and sex of mice, the immunity to the same antigen is different. 3. Operation errors and stress to mice occurred during the injection process. We will verify this problem through follow-up experiments.
Point 8: The authors discussed the effect of sgRNA GC content on editing efficiency, but authors should illustrate the GC content of the sgRNA you designed and discuss how it affect efficiency in the present study.
Response 8: Thanks to the reviewer for the suggestion. In this study, the starting bases of the sgRNA we designed were mainly thymine (T), cytosine (C) or guanine (G), and the CG content was controlled as much as possible to about 50%. It has been reported that sgRNAs starting from thymine, guanine or cytosine induce relatively high cleavage efficiency. Furthermore, sgRNAs with high AT content induce higher cleavage efficiency than sgRNAs with high GC content, which may be due to the fact that more energy is required to unravel the three hydrogen bonds between guanine and cytosine in the DNA double helix structure[2]. It has been stated in the Discussion section as suggested by the reviewer (We marked it in red on page 15, lines 488-495).
- Zeng B, Zhan S, Wang Y, Huang Y, Xu J, Liu Q, Li Z, Huang Y, Tan A. Expansion of CRISPR targeting sites in Bombyx mori. Insect Biochem Mol Biol. 2016 May;72:31-40. doi: 10.1016/j.ibmb.2016.03.006. Epub 2016 Mar 24. PMID: 27032928.
Point 9: At page 15 line 481, the authors mentioned that the proportion of fluorescent cells, please provide the proportion of each generation as supplementary materials.
Response 9: According to your suggestion, we uploaded the proportion of fluorescent cells.
Point 10: At page 16 line 515, NHEJ is the main method of DSB repair, not HR.
Response10: It has been revised as suggested by the reviewer(We marked it in red on page 17, lines 575).
Point 11: Please cite some latest references.
Response11: It has been revised as suggested by the reviewer. Replaced references 1, 2, 4, 5, 6, 7, 11, 12, 13, 20, 37, 41, 42.

Reviewer 3 Report
Dear authors,
In this paper, the authors try to generate different BoHV-1 using the CRISP/Cas9 system. The manuscript is well written and structured, the introduction provides sufficient background, the research design is appropriate, and the results are clearly presented. However, in the statistical analysis, the authors should modify certain information. Mathematically, very significant does not exist. The authors must stick to the p-value <0.05 that they have estimated when performing the ANOVA test. They should remove p<0.01 and p<0.001 from the entire manuscript. In addition, before carrying out a parametric test (ANOVA) they must check the normality and homoscedasticity of the data.
On the other hand, the results and discussion do not clearly indicate the purpose of the work and its application.
Author Response
Response to Reviewer 3 Comments
Point 1: In this paper, the authors try to generate different BoHV-1 using the CRISP/Cas9 system. The manuscript is well written and structured, the introduction provides sufficient background, the research design is appropriate, and the results are clearly presented. However, in the statistical analysis, the authors should modify certain information. Mathematically, very significant does not exist. The authors must stick to the p-value <0.05 that they have estimated when performing the ANOVA test. They should remove p<0.01 and p<0.001 from the entire manuscript. In addition, before carrying out a parametric test (ANOVA) they must check the normality and homoscedasticity of the data.
Response 1: The rigorous attitude of reviewers to the statistical analysis of scientific research data deserves our study. We apologize for not describing in detail the statistical methods used. In fact, before conducting a statistical test, we computed the variance and normality in the sample. For samples that lacked normality, we used Wilcoxon Rank Sum Test. For samples that conform to normality, we employed an ANOVA 1-way analysis with Tukey Post-Hoc test for individual comparisons, when equal variance was satisfied. In cases where equal variance was not satisfied, we used the Kruskal-Wallis test.
According to the reviewer’s comment, relevant information has been supplemented in “2.10 Statistical analyses”(We marked it in red on page 6, lines 225-230). And according to the reviewer's comments, p<0.01 and p<0.001 were deleted, and very significant was changed to significant.
Point 2: On the other hand, the results and discussion do not clearly indicate the purpose of the work and its application.
Response 2: The purpose of this study is to establish a CRISPR/Cas9 method to rapidly knock out the gE and US9 genes of BoHV-1 at the same time, and to recombine the antigenic dominant proteins of BVDV or other viruses, so that BoHV-1 becomes a delivery vehicle for subunit vaccines. This study provides a preliminary experimental basis for the development of future BoHV-1 gene deletion vaccines and genetically engineered subunit vaccines. At the same time, according to the results of this study, we found that the non-coding region between the gE and US9 genes has a certain effect on BoHV-1, which can prevent us from knocking out the non-coding region between the BoHV-1 gene in our subsequent experiments.
Additional information has been added in lines 277-279 and364-368 of the results section and lines 465-473 of the discussion section based on the reviewer's comment(We marked with red font).

Round 2
Reviewer 3 Report
The authors have made the indicated corrections and the manuscript can be published in its current form.
Author Response
请参阅附件。
